# Risk of Hypertension and Use of Antihypertensive Drugs in the Physically Active Population under-70 Years Old—Spanish Health Survey

**DOI:** 10.3390/healthcare10071283

**Published:** 2022-07-11

**Authors:** Ángel Denche-Zamorano, Jorge Pérez-Gómez, Maria Mendoza-Muñoz, Jorge Carlos-Vivas, Rafael Oliveira, João Paulo Brito

**Affiliations:** 1Promoting a Healthy Society Research Group (PHeSO), Faculty of Sport Sciences, University of Extremadura, 10003 Cáceres, Spain; andeza04@alumnos.unex.es (Á.D.-Z.); jorgecv@unex.es (J.C.-V.); 2Health Economy Motricity and Education (HEME) Research Group, Faculty of Sport Science, University of Extremadura, 10003 Cáceres, Spain; jorgepg100@unex.es; 3Research Group on Physical and Health Literacy and Health-Related Quality of Life (PHYQOL), Faculty of Sport Sciences, University of Extremadura, 10003 Cáceres, Spain; 4Departamento de Desporto e Saúde, Escola de Saúde e Desenvolvimento Humano, Universidade de Évora, 7004-516 Évora, Portugal; 5Sports Science School of Rio Maior, Polytechnic Institute of Santarém, 2040-413 Rio Maior, Portugal; rafaeloliveira@esdrm.ipsantarem.pt; 6Research Center in Sport Sciences, Health Sciences and Human Development, Quinta de Prados, Edifício Ciências de Desporto, 5001-801 Vila Real, Portugal; 7Life Quality Research Centre, 2040-403 Rio Maior, Portugal

**Keywords:** blood pressure, health, medication, physical fitness, sedentary lifestyle

## Abstract

Introduction: Reducing the prevalence of hypertension is a major priority of the World Health Organization (WHO). Its high prevalence and associated risks generate high economic and social costs. Physical activity (PA) is associated with a decrease in hypertension and in the use of antihypertensive drugs. Objective: To explore the association between PA levels (PAL), prevalence of hypertension and the use of antihypertensive drugs in Spanish population. To calculate risks of hypertension and use of antihypertensive in the inactive versus physically active population. Method: This cross-sectional study used data from 17717 individuals, sampled in the 2017 National Health Survey. Interaction by sex, age groups, body mass index (BMI), hypertension prevalence, antihypertensive drugs use and PAL, using a pairwise z-test, and dependence relationships between variables, were studied using a chi square test. Odds ratios of hypertension and antihypertensive drug use were calculated among the inactive and the physically active populations. Results: The findings showed a significant inverse association between prevalence of hypertension, antihypertensive use, and PAL in both sexes and different age and BMI groups, with lower prevalence of hypertension and antihypertensive use when PAL were higher. The risks of hypertension and antihypertensive use seems to be reduced when related to higher PAL compared to inactive people. Conclusions: High PAL is associated with lower prevalence of hypertension and lower antihypertensive use. Thus, being physically active or very active may reduce the risks of suffering from hypertension and the need to use antihypertensives compared to inactive people or walkers.

## 1. Introduction

Around 1.5 billion people worldwide have hypertension, with a prevalence that reaches a third of the adult population, causing 8 million deaths a year [1]. High blood pressure (BP) is a leading cause of death and disability, causing 13.5% of the world’s premature deaths and 6% of its disabilities [2].

Although hypertension is an important and common risk factor for cardiovascular disease [3], it is not always taken seriously and is often poorly controlled [4]. The lack of symptoms and the fact that people may not be aware that they have high BP makes this disease a marked health risk [5]. The high prevalence of this condition is a great concern to all governments and world health organizations [6], since it has been associated with multiple comorbidities and cardiovascular diseases [7] and represent one of the greatest risk factors for all-cause mortality [8]. As an example, half of all strokes and ischemic heart diseases can be attributed to high BP [9]. Obesity is another major risk factor for hypertension that has reached pandemic proportions [10], and research shows that in around two-thirds of people with hypertension the condition is directly attributable to obesity [11,12].

In European countries, the prevalence of hypertension in the adult population ranges from 30% to 50% [13]. The prevalence of hypertension with metabolic syndrome in the general population of Spain was 11% in 2010 [14]. Several studies also reported that one reason for the high prevalence of hypertension and the use of pharmacological therapies (antihypertensive) is the aging of the population but also the habits of current societies: smoking, diet, stress, the presence of diabetes, chronic kidney disease, being overweight or obese, genetic components and family history as well as physical inactivity [15,16,17,18]. Although there is increasing emphasis on treatment by drugs, lifestyle modification is an important part of hypertension management [19,20].

The American College of Cardiology (ACC)/American Heart Association (AHA) Task Force on Clinical Practice Guidelines recently redefined hypertension to a lower systolic BP (SBP) threshold of 130 mmHg or 80 mmHg for Diastolic BP (DBP) [21], while the American Joint National Committee Seven defined a threshold of 140 mmHg for SBP or 90 mmHg for DBP [22]. A person is considered to have hypertension when the SBP is ≥130 mmHg and the DBP ≥80 mmHg, measured on two different days [23]. Although the previous diagnosis is not unique, there are also different categories for BP values, such as high BP (SBP between 120 and 129 mmHg and DBP less than 80 mmHg) and stage 1 hypertension (SBP between 130 and 139 mmHg and DBP between 80 and 89 mmHg), as well as hypertensive crises, with SBP greater than 180 mmHg and DBP higher than 120 mmHg, which represent a vital emergency [21,24].

This change underscores the importance of hypertension as a public health problem. The authors of the recent ACC/AHA guidelines stated that nearly all of those newly diagnosed with hypertension, due to the lower BP threshold, can treat their hypertension with lifestyle modification rather than drugs [21]. They also emphasized that decreasing the prevalence and improving the control of hypertension by increasing the use of lifestyle antihypertensive therapy, such as participation in habitual physical activity (PA), would provide major societal public health and economic benefit [21].

Common drugs for the treatment of hypertension are called antihypertensive. Patients who had both SBP higher than 140 mmHg and DBP higher than than 90 mmHg usually take antihypertensive drugs to control hypertension. However, there are several drugs that aim to reduce BP, the choice of which depends on BP range, age, or general health, and can be combined with each other to increase effectiveness. The most commonly used antihypertensives are diuretics, angiotensin-converting enzyme inhibitors, angiotensin II receptor antagonists, calcium channel blockers, alpha-blockers, beta-blockers, and aldosterone antagonists, among others [25,26]. The consumption of these drugs have some side effects that can reduce treatment adherence, with excessive diuresis or incontinence problems being some of the main effects, along with cough, dizziness or skin lesions [27,28]. The cost of these drugs represents a remarkably high expense for health services. In the US, an annual cost of USD 336 per hypertensive patient is estimated [29].

PA could be a therapy to prevent or control hypertension. Some research found an inverse relationships between PA and the incidence of hypertension, as well as reductions in BP in people with normal BP, prehypertension and hypertension by engaging exercise training programs [30,31].

Lifestyle modification should be an important part of therapy when hypertension is first diagnosed, with or without starting antihypertensive drugs [32]. However, there is no clear evidence about certain factors, such as sex, age and weight, as well as the frequency, type and intensity of PA that may influence the associations between PA and BP [30,31]. For all these reasons, in addition to pharmacological treatment, control of unhealthy habits is recommended, reducing risk factors, such as tobacco, alcohol, stress, hypercaloric diets and physical inactivity, among others, and promoting healthy lifestyle habits that favour, prevent or reduce hypertension and/or high BP values [4,33].

There is scarce information about the prevalence of hypertension and the use of hypertensive drugs in the Spanish population, according to sex, age, body mass index (BMI) or PA level (PAL). Thus, the aim of this study was twofold: (i) to analyse the relationship between the prevalence of hypertension and the use of antihypertensive drugs in different population groups according to sex age group, BMI and PAL; (ii) to determine the probability of the risk of suffering from hypertension and the use of antihypertensive drugs according to the PAL. The hypotheses of this study were that the prevalence of hypertension and use of antihypertensives drugs is associated with sex, age group, BMI and PAL and that being physically active reduces the risks of suffering from hypertension and antihypertensive drugs use compared to the inactive population or walkers.

## 2. Materials and Methods

### 2.1. Design

This study uses a prevalence-based approach that combined the demographics of the population with hypertension and antihypertension drugs, incidence of obesity, PAL and healthcare costs. This research consisted of a cross-sectional study with data from the Spanish National Health Survey 2017 (SNHS2017/ENSE2017) [34]. This survey was the only major survey carried out in Spain on the health of the population, prior to the COVID-19 pandemic. The SNHS is a survey carried out by the Ministry of Health, Consumption and Social Welfare (MHCSW), together with the National Institute of Statistics of Spain, every five years, with the aim of finding out the state of health of residents. The SNHS2017 was carried out between October 2016 and October 2017. The MHCSW instructed and accredited the personal interviewers who conducted the surveys in person with the previously selected participants with a stratified triphasic random sampling system [35]. Before receiving the interviewers, the MHCSW sent a letter to the participants, communicating their inclusion in the survey, the request to participate, as well as the reasons for it and the anonymous and confidential treatment during the treatment and exposure of the data, as well as the regulations governing the protection of said data.

### 2.2. Subjects

The initial sample was based on the 23,089 respondents who completed the SNHS2017, all residents in Spain and aged between 15 and 103 years. The following inclusion criteria were established: being equal or over than 15 years old, presenting all the data in the items referring to BP (question, Q.113–Q.117 in the SNHS2017), sex and age, as they were requested in the SNHS2017 Adult Questionnaire [36]. Those over than 70 years old were excluded, because they were not questioned about their BP in the SNHS2017 (5312 participants) and people who did not present values in all the items referring to BP or answered “Do not know” or “No answer” (60 participants). The resulting sample was 17,717 participants. For analysis that included data on the prevalence of hypertension and drugs use for hypertension, participants without data on item Q.25a.1 (21 participants) were excluded, resulting in a sample of 17,696 participants. In the analysis that included the use of drugs, participants who did not present data in the items Q.85 and Q.87a.7 (2 participants) were excluded, resulting in a sample of 17,694 participants. For analysis that included BMI, all participants without data on the body weight and height item (498 participants) were excluded. The classification of blood pressure for adults used to classify the BP values was: normal blood pressure values considered to be below 120/80 mmHg, 120–139/80–89 mmHg is considered high normal and values above 140–90 mmHg are considered hypertension [36].

### 2.3. Procedures

Data were extracted from public files provided by the MHCSW. The variables used are described in Table 1.

### 2.4. Statistical Analysis

The normal distribution of all variables was tested using the Kolmogorov–Smirnov test. Subsequently, a descriptive analysis was carried out, presenting the prevalence of hypertension and use of hypertensive drugs according to the different groups: general population, sex, age and BMI groups and PAL by absolute and relative frequencies. An analysis of dependence was performed between these sociodemographic biases studied (chi square test) and contrast of proportions between the different groups (paired z-test for independent proportions), as well as these same prevalence between these prevalence of hypertension and use of antihypertensive drugs according to the PAL for all sociodemographic groups performed. Effect sizes are reported: Phi (φ) for 2 × 2 contingency table and Cramer’s V (V) for a contingency table larger than 2 × 2 being interpreted according to Lee [39]. Odds ratios (ORs) were calculated accordingly with corresponding 95% confidence intervals (CIs) of the risk of suffering from hypertension and using antihypertensive drugs between the different PALs, taking the inactive group as a reference out of all the analysed population groups. A multiple binary logistic model was used to study effects of predictor variables (Age, BMI and PAL) on hypertension status and use of antihypertensives. Two-sided *p*-values ≤ 0.05 were considered statistically significant.

Statistical analyses were applied using IBM SPSS Statistics for Windows, Version 25.0 (IBM Corp., Armonk, NY, USA).

## 3. Results

Table 2 presents a descriptive analysis on the prevalence of hypertension in the general population, by sex, age groups, BMI and PAL groups. Dependency relationships were found between the prevalence of hypertension and all the sociodemographic variables (*p* < 0.001 in the chi square area). Significant differences were found between the proportions of hypertension between men and women, as well as in all age groups and BMI (*p* < 0.05 in the z-test). However, no differences were found in the proportions of hypertensive patients between Inactive and Walkers, but there were between the Active and Very Active groups (*p* < 0.05).

Table 3 shows the prevalence of antihypertensive drug use in the entire population and the different sociodemographic biases. Dependency relationships were found between the use of antihypertensive drugs and sex, age groups, BMI and PAL (*p* < 0.001 in the chi square test). Significant differences were found in the proportions of use of antihypertensive drugs between men and women, the different age groups and BMI (*p* < 0.05 in the z-test). However, no differences were observed in these proportions between Inactive and Walkers, although there were differences between PAL and the Active and Very Active levels (*p* < 0.05).

The dependence analysis was performed between the prevalence of hypertension and PAL, finding significant associations in both sex (*p* < 0.001), in Young (*p* < 0.05), Young Adults and Older Adults (*p* < 0.001) and Older (*p* < 0.05) groups. These relationships were not found in people with a BMI lower than 18.5 (*p* = 0.083), although they were found in the rest of the groups (*p* < 0.001).

Table 4 also shows the analysis of the proportions of the prevalence of hypertension, according to PAL, for all population groups.

Table 5 shows the prevalence of using antihypertensive drugs, according to PAL, in different population groups, finding relationships in men and women (*p* < 0.001) and in all age groups. No dependence relationships were found between the use of antihypertensive drugs and PAL in the group with a BMI < 18.5 (*p* = 0.228), although they were found in the other BMI groups (*p* < 0.001). Table 4 also shows the differences in the proportions of antihypertensive use, according to PAL, in all population groups.

Table 6 shows the risks of suffering from hypertension, according to PAL, taking as a reference the prevalence of hypertension in Inactive people for all population groups analysed. Significantly lower risks of hypertension were found in Active and Very Active compared to Inactive.

Table 7 shows the probability risks of antihypertensive use, regarding PAL, for all the population groups analysed, taking as reference the prevalence of antihypertensive use in Inactive group. The Active and Very Active groups show significant values associated with reduced risks compared to the Inactive in all population groups with the exception of participants with BMI < 18.5.

The data presented in Figure 1 show that no differences were found in the prevalence between Active and Very Active women for hypertension and use of antihypertensive drugs; however, in men there is a decrease of 6.1 and 6.4 percent points, respectively.

Figure 2 shows a reduction in the prevalence of hypertension and antihypertensive use with the increase in the PAL in people with different BMI categories (except for BMI < 18.5).

In the general population, the lowest risk of hypertension was found in the Very Active group (OR: 0.41, 95% CI: 0.35–0.49). The same was found in men (OR: 0.36, 95% CI: 0.29–0.45), women (OR: 0.46, 95% CI: 0.35–0.61) (Figure 3). These reduced risk values were found in other population groups, such as Young, Young Adults, as well as people with normal weight, overweight or obesity. In Older and Older adults, the lowest risk was found in the Active group. No significant OR were found in people with BMI < 18.5.

In the general population the lowest risk of antihypertensive drugs used was found in Very Active people (OR: 0.32, 95% CI: 0.27–0.39). Lower values of reduced risks of antihypertensive use were found in all almost PALs (except the Walkers and Very Active older). When analysed by age groups, the risk of antihypertensive drugs use was lower in Active and Very Active population, except for the Very Active Older individuals.

Based on the results of binary regression analyses regarding hypertension status and antihypertensive use, those who were young, female, normal weight and physically very active showed lower risks of hypertension and antihypertensive use. These models explained 30.1% and 36.4% (Nagelkerke R2) of the variance in hypertension status and antihypertensive use, respectively (Table 8).

## 4. Discussion

The aim of this study was to analyse the relationship between the prevalence of hypertension and the use of antihypertensive drugs in different population groups according to sex, age group, BMI and PAL. The study also aimed to determine the probability of the risk of suffering from hypertension and using antihypertensive drugs according to the PAL. In this sense, dependency relationships were found between the prevalence of hypertension and the different sociodemographic variables analysed: sex, age, BMI and PAL.

The prevalence of self-reported hypertension in the general adult population in Spain, before the arrival of the COVID-19 pandemic, was 18.2%, being higher in men than in women (20.2% vs. 16.4%, *p* < 0.001, ESφ = 0.049); however, the study by Menendez et al. reported values of around 50% in men and slightly less than 40% in women [8]. The ranges found in the present study are lower than the 30% estimated prevalence in the world population [23]; however, there is research that suggests that 50% of hypertensive patients are unaware of this pathological condition [5], which would place this prevalence within global ranges. Thoenes et al. (2010), in a large international survey conducted in 26 countries, reported BP-control rates in the population with or without diabetes mellitus, in men and women, respectively, of 37.7% (36.69–38.75%) and 34% (32.92–35.06%) in SBP and of 63.6% (62.61–64.66%) and 64.1% (63.01–65.19%) in DBP. Another study in Brazil [40] showed that the prevalence of self-reported high BP was 20.9%, with 21% in urban areas and 20.1% in rural areas (OR = 1.06). In both areas of the Brazil’s study, the likelihood of reporting high BP increased with age and in women [40].

In the European countries, the prevalence of hypertension in the adult population ranges from 30% to 50%, and it will likely increase as a consequence of population ageing and current lifestyles [13,41]. The sex difference worldwide became significant at an age higher than 45–49 years [42]. In the present study, the prevalence was found to increase with higher age and BMI values: age (2.7% Younger vs. 47.5% Older) and BMI (3.1% Underweight vs. 38.5% Obese). These same dependency relationships and differences in proportions were found in the use of antihypertensive drugs, being higher in men than in women (16.5% vs. 12.9%), in older people than in Young people (45.2% vs. 0.5%) and in people with Obesity than in those Underweight (32.8% vs. 1.9%).

Furthermore, BP-control rates in women with known cardiovascular disease or cardiovascular risk factors appear to be lower at a higher age compared with men [43,44,45]. As cardiovascular disease is considered to be the single largest cause of mortality in women with 70% of cardiovascular deaths attributable to modifiable risk factors, such as hypertension, it is important to better understand sex disparities in hypertension management to ensure equal standards in treatment for both men and women [46].

In the large survey conducted by Thoenes et al., in countries in Europe, North America, Latin America, Middle East and Asia as well as in Turkey, Australia and Morocco, the number of antihypertensive drugs used in men and women in the overall population was 30.1% (29.12–31.04%) of men and 30.9% (29.84–31.90%) of women was one drug, while 40% (38.95–41.00%) of men and 39.4% (38.32–40.50%) of women used two drugs, and 29.9% (28.98–30.90%) of men and 29.7% (28.70–30.74%) of women used three or more drugs [42]. Regarding the mean number of drugs used, Thoenes et al. reported that no major sex disparities were observed [42]. In our study, there were differences between men and women in the use of antihypertensive drugs; however, the effect size is very small (*p* < 0.001, ESφ = 0.051).

Both antihypertensive pharmacological intervention and PA intervention have been recommended as first-line treatments for hypertension [47]. An extensive body of research suggested PA may have beneficial effects for people with hypertension [47,48,49].

When analysing the use of antihypertensive drugs in relation to the PAL, no differences were found between men and women in the groups of Inactive and Walkers and between these two groups for the entire population. Only in the Very Active group are differences between men and women. The Active and Very Active groups differ from the other groups in the prevalence of drugs use. According to Pescatello et al., the PA effects on the prevalence and use of antihypertensive drugs should be analysed through the study of the BP response to an exercise training intervention ranging from low to vigorous intensity [30]. Several meta-analyses of a randomized controlled trial examined the BP response to an exercise training intervention ranging from low to vigorous intensity among adults who were physically inactive at baseline and with hypertension [48,50,51,52,53,54,55,56,57,58,59,60,61,62]. They reported a significant reduction in BP. The magnitude of the BP reductions ranged from 5 to 17 mmHg for SBP and 2 to 10 mmHg for DBP. There seems to be strong evidence that low to vigorous PA reduces BP among adults with hypertension. In a meta-analysis by Pescatello et al., an association is reported between high amounts (i.e., volume and/or intensity) of leisure-time PA with a 19% decreased risk of incident hypertension compared to the group that engaged in low amounts of leisure-time PA (relative risk (RR) = 0.81, 95% CI: 0.76–0.85) [30]. It is further noted in Pescatello et al., that moderate amounts of leisure-time PA was associated with an 11% decreased risk of hypertension compared to the group that engaged in low amounts of leisure-time PA (RR, 0.89, 95% CI: 0.85–0.94) [30]. According to Bakker et al., even small amounts of PA seems to be related to lower risk of hypertension, whereas higher levels of PA and cardiorespiratory fitness could promote a decrease to the risk of hypertension [49]. Pescatello et al. concluded that PA seems to be related to BP reduction among adults with prehypertension and hypertension [30].

However, there is a dose–response relationship between the PAL and incident hypertension [62,63]. Liu et al. quantified the dose–response relationship between PA and incident hypertension (BMI adjustment as a covariate) and found it to be linear. Although most dose–response studies are based on self-reported PA data measured by self-reported questionnaires, there seems to be strong evidence on the inverse dose–response relationship between PA and incident hypertension [63]. The findings of our study show that in the Active (13.5%) or Very Active (10.0%) group there was significant lower prevalence (*p* < 0.05 in z-test) compared to Inactive and Walkers (21.2% and 22.4%, respectively). In fact, the use of antihypertensive was lower in the Very Active group (6.6%) than in Active group (10.6%), and than in Inactive people and Walkers (17.9% and 18.3%, respectively).

PA seems to be related to the prevalence of hypertension and antihypertensive use, both in men and women, compared to Inactive and Walkers. This has already been found in other studies [49]. Williams et al. found that adult runners had a 38% lower risk of hypertension compared to walkers [64]. In our study, no differences were found in the prevalence for hypertension and use of antihypertensive between Active and Very Active women but in men there was a decrease of 12.8% and 6.4%, respectively, as can be seen in Figure 1.

PA seems to contribute to lower values of the prevalence of hypertension through its acute effects, with particular relevance in the population identified with prehypertension, which considered a critical population [65]. Some investigations have studied the acute response on BP reduction [66,67] because they considered it relevant to the chronic effects [66].

Lifestyle modification, including the increase of PA, is the only recommended treatment in the prehypertension phase [22]. Thus, the observation of PA for prehypertensive subjects is warranted [67].

The review study by Pescatello et al. reported that PA reduced BP normotensive, prehypertension and hypertension adults [30]. However, randomized controlled trials with longer intervention periods and observational studies with objectively measured PA are needed to define the exact dose–response relationship of PA and hypertension.

Several studies [68,69,70,71,72] stated that the parameter with the greatest association with hypertension reduction is cardiorespiratory fitness; however, it is essential to adjust for potential confounding factors, such as body fat percentage and BMI at baseline.

Despite the data in the present study being self-reported in groups with different BMI categories, increasing PA level also seemed to reduce the prevalence of hypertension and antihypertensive medication use. This was found in all the groups, according to the BMI categories, except in the underweight group (BMI < 18.5), probably due to the small sample size (Figure 2). In overweight (BMI, 25–30) and obese groups (BMI ≥ 30), an increase in PAL could help to reduce BP, as reported in other studies [73,74].

Our findings suggest an association between the prevalence of hypertension and antihypertensive medication use and PAL in both sexes in different age groups and BMI groups, with lower magnitudes found in higher levels of PA. The risks of hypertension and antihypertensive use seem to be lower when associated with higher levels of PA compared to Inactive people. A higher PAL is related to a lower prevalence of hypertension and a lower prevalence of antihypertensive use. To be physically Active or Very Active seems to be associated with lower risks for prevalence of hypertension and use of antihypertensive compared to being Inactive and Walkers in the Spanish population.

### 4.1. Practical Applications

The findings of this study could be used by public organisations, as well as by national and regional health services, applying public health policies to the promotion of health through PA. It would be advisable to invest in health education throughout a person’s life cycle, including sports science professionals in health services, promoting the inclusion of active breaks in educational and work centres, creating PA programs for the elderly, creating multidisciplinary teams in health prevention and in the treatment of hypertension.

The results of this research could be used as a comparative reference framework for future research with similar data from different points during the COVID-19 pandemic or post-pandemic research once the post-pandemic National Health Surveys have been published.

### 4.2. Limitations

The current research has some limitations as the study is based on a national epidemiological survey that evaluated the effects of a broad spectrum of PA on BP and the incidence of hypertension and antihypertensive medication in the Spanish population under 70 years old. The study is based on self-reported on PA data. Some survey questions have similarities and are sometimes not closed-ended questions. Subjectively measured PA is often overestimated, which results in an underestimation of the benefits of PA on health and may ignore potential health benefits of lower PAL.

The types of antihypertensive drug and the dosage used by the respondents were not included in the survey questions. These factors may have an influence on the results obtained in the data analysis.

Causal relationships cannot be established, given the type of design, and it would be convenient to delve into the results found using designs with which cause–effect relationships could be established, where exposure to PA and BP response would be analysed separately.

Although different sociodemographic biases have been considered, there are many others that have not: educational level, diet and purchasing power, among others. It was not possible to know the medical history of the participants, and it was not possible to obtain objective data on PA performed by the participants. It would be convenient to use other methodologies in future National Health Surveys in Spain, including parameters and medical follow-ups, as well as PA, using low-cost inertial devices or mobile applications.

We summarized relevant previous research and the evidence gaps around the effects of BP reduction in the very young and the very old and whether the recommended use of age-dependent BP thresholds for the initiation or intensification of treatment by some clinical practice guidelines [75,76] is justified.

Nonetheless, providing reliable and precise quantification of expected treatment effects in a wide range of at-risk groups [77] is an important and necessary step towards providing better decisions and addressing those implementation gaps.

## 5. Conclusions

This research shows that a high PAL is associated with lower prevalence of hypertension and antihypertensive use, both in the general population and in men and women of different ages and BMI. PA seems to be related to a reduced BP in normotensive, prehypertension and hypertension adults.

As stated in the review by Pescatello et al. [30], being physically active or very active appears to be associated with lower risks of the prevalence of hypertension and antihypertensive use compared with being inactive or a walker. Performing moderate and/or vigorous PA appears to have a positive impact on predisposition to the risks of hypertension, as well as the need to consume antihypertensives. Nonetheless, controlled trials with longitudinal interventions and randomized observational studies with objective measures of PA are needed to characterize the exact dose–response relationship between PA and hypertension.

## Figures and Tables

**Figure 1 healthcare-10-01283-f001:**
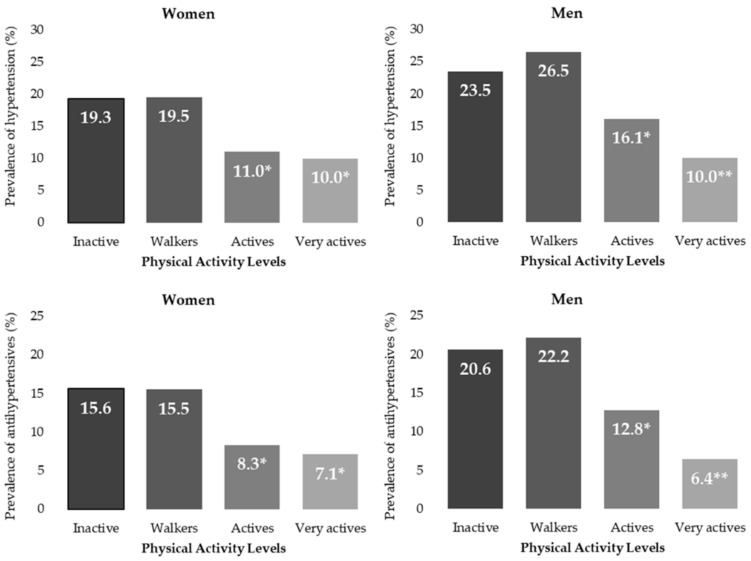
Prevalence of hypertension and use of antihypertensive drugs by PAL and sex. */** (One star shows significant differences in prevalence of hypertension/antihypertensives with respect to no star and two star values. Two stars show significant differences with no star and one star values. *p* < 0.05 from z-test.

**Figure 2 healthcare-10-01283-f002:**
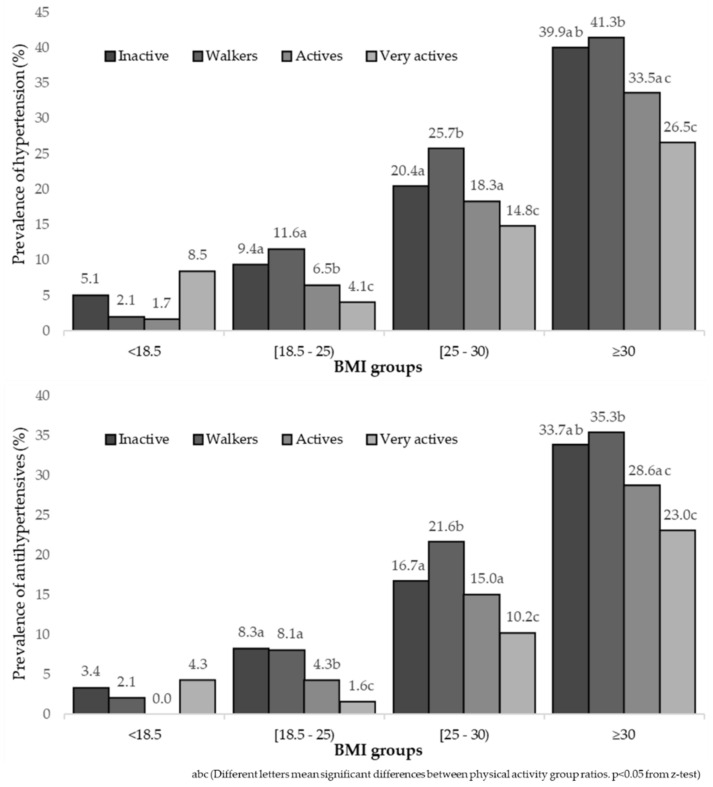
Prevalence of hypertension and the use of antihypertensive drugs, according to the level of physical activity by body mass index (BMI) groups.

**Figure 3 healthcare-10-01283-f003:**
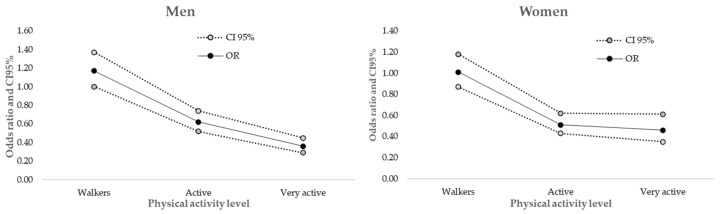
Risk of hypertension, according to physical activity level by sex.

**Table 1 healthcare-10-01283-t001:** Description of variables.

Variables	Description
Sex	Male or female
Body mass index (BMI) groups	Grouped as underweight (<18.5 kg/m^2^), normal weight (18.5 to 24.9 kg/m^2^), overweight (25 to 29.9 kg/m^2^), obese (≥30 kg/m^2^).
Hypertension status	Extracted item Q.25a.1 “Do you suffer, or have you ever suffered from hypertension?”. Answer: “Yes”, “No”, “I don’t know” or “Don’t answer”.
Age groups	The participants were grouped in the following age ranges: Young (15–34 years), Young adults (35–49 years), Older adults (50–64 years) and Older (65–69 years).
Use of antihypertensive drugs	Obtained from answers to items Q.85 “During the past 2 weeks, have you taken any drugs prescribed to you by a doctor?” Answer: “Yes” or “No”; and Q.87a.12 “In the following, I will read you a list of types of drugs, please tell which one(s) you have taken in the last 2 weeks, and which ones have been prescribed to you by the doctor: Blood pressure medicines?” Answers: “Yes”, “No”, “I don’t know” or “Don’t answer”. Participants were grouped into: Yes (who answered “yes” to items Q.85 and Q.85a.12) or No (who answered “no” to item Q.85, and who answered “yes” to item Q.85 and “no” to item Q.87a.12.
Physical activity level (PAL)	This variable grouped the participants according to the PA reported in the SNHS2017. For this purpose, a PA index (PAI) was created with the answers given to items Q.113–Q.117 (Q.113 “First of all, think about the intense activities you have done in the last 7 days. Intense activities are those that require a lot of physical exertion and make you breathe much harder than normal, such as heavy lifting, digging, aerobic exercise or fast cycling. Think only of those that you did for at least 10 min at a time.”—“Days per week of Vigorous PA”; Q.114 “Average duration of Vigorous PA performed in a week”; Q.115 “Now please think of all those moderate activities you have done in the last 7 days. Moderate activities are those that require moderate physical exertion that makes you breathe a little harder than normal, such as carrying light weights, cycling at regular speed or playing doubles tennis. Think only of those you did for at least 10 min at a time.”—“Days per week of Moderate PA”; Q.116 “Average duration of Moderate PA performed in a week” and Q.117 “Days in a week that they walk, less than 10 min in a row”), this PAI was described by previous studies [37,38]. The PAI could take values between 0 and 67.5, the groups were as follows: Inactive (PAI = 0 and Q.117 = 0); Walkers (PAI = 0 and Q.117 > 0); Active (PAI between 1 and 30) and Very Active (PAI > 30).

**Table 2 healthcare-10-01283-t002:** Population characteristics by hypertension status in Spanish 15–69-year-olds in SNHS2017 (data presented in absolute and relative values).

Characteristic	Hypertension		No Hypertension		*χ* ^2^	df	*p*	ES
Overall	3228	(18.2%)	14,468	(81.8%)	n.a.	n.a.	n.a.	n.a.
Sex								φ
Men	1713	(20.2%) ^a^	6762	(79.8%)	42.4	1	<0.001	0.049
Women	1515	(16.4%) ^b^	7706	(83.6%)				
Age (years)					*χ* ^2^	df	*p*	V
15–34	103	(2.7%) ^a^	3764	(97.3%)	2436.0	3	<0.001	0.371
35–49	569	(9.2%) ^b^	5600	(90.8%)				
50–64	1744	(29.3%) ^c^	4207	(70.7%)				
65–69	812	(47.5%) ^d^	897	(52.5%)				
BMI (kg/m^2^)					*χ* ^2^	df	*p*	V
<18.5	13	(3.1%) ^a^	401	(96.9%)	1390.8	3	<0.001	0.284
[18.5–25)	664	(8.6%) ^b^	7092	(91.4%)				
[25–30)	1340	(21.6%) ^c^	4850	(78.4%)				
≥30	1094	(38.5%) ^d^	1746	(61.5%)				
PAL					*χ* ^2^	df	*p*	V
Inactive	538	(21.2%) ^a^	1994	(78.8%)	285.4	3	<0.001	0.127
Walkers	1808	(22.4%) ^a^	6250	(77.6%)				
Active	661	(13.5%) ^b^	4226	(86.5%)				
Very Active	221	(10.0%) ^c^	1998	(90.0%)				

BMI, body mass index; PAL, physical activity level; n.a., not applicable; *χ*^2^, Pearson’s chi square; df, degree of freedom; *p*, *p*-value; ES, effect size; V, Cramer’s V; φ, Phi’s coefficient; ^a,b,c,d^, each subscripted letter indicates differences between the population’s characteristics with hypertension diagnosis, at 95% z-test for independent proportions.

**Table 3 healthcare-10-01283-t003:** Population characteristics, according to the use of antihypertensives in Spanish 15–69-year-olds in SNHS2017.

Characteristic	Antihypertensives		No-Antihypertensives		*χ* ^2^	df	*p*	ES
Overall	2591	(14.6%)	15,104	(85.4%)	n.a.	n.a.	n.a.	n.a.
Sex								φ
Men	1400 ^a^	(16.5%)	7074	(83.5%)	45.9	1	<0.001	0.051
Women	1191 ^b^	(12.9%)	8030	(87.1%)				
Age (years)					*χ* ^2^	df	*p*	V
15–34	18 ^a^	(0.5%)	3849	(99.5%)	2831.3	3	<0.001	0.400
35–49	322 ^b^	(5.2%)	5847	(94.8%)				
50–64	1479 ^c^	(24.9%)	4471	(75.1%)				
65–69	772 ^d^	(45.2%)	937	(54.8%)				
BMI (kg/m^2^)					*χ* ^2^	df	*p*	V
<18.5	8 ^a^	(1.9%)	406	(98.1%)	1334.8	3	<0.001	0.279
[18.5–25)	458 ^b^	(5.9%)	7297	(94.1%)				
[25–30)	1099 ^c^	(17.8%)	5091	(82.2%)				
≥30	932 ^d^	(32.8%)	1908	(67.2%)				
PAL					*χ* ^2^	df	*p*	V
Inactive	454 ^a^	(17.9%)	2078	(82.1%)	287.2	3	<0.001	0.127
Walkers	1474 ^a^	(18.3%)	6583	(81.7%)				
Active	516 ^b^	(10.6%)	4371	(89.4%)				
Very Active	147 ^c^	(6.6%)	2072	(93.4%)				

BMI, body mass index; PAL, physical activity level; n.a., not applicable; *χ*^2^, Pearson’s chi square; df, degree of freedom; *p*, *p*-value; ES, effect size; V, Cramer’s V; φ, Phi’s coefficient; ^a,b,c,d^, each different subscripted letter indicates indicate significant differences between the proportions of people using antihypertensives, according to sex, age, BMI and physical activity at 95% z-test for independent proportions.

**Table 4 healthcare-10-01283-t004:** Prevalence of hypertension by physical activity level and: sex, age and BMI.

	Physical Activity Levels											
Variables	Inactive		Walkers		Active		Very Active					ES
Sex	*n* (%)		*n* (%)		*n* (%)		*n* (%)		*χ* ^2^	df	*p*	φ
Male	276 ^a^	23.5%	897 ^a^	26.5%	395 ^b^	16.1%	145 ^c^	10.0%	212.2	3	<0.001	0.158
Female	262 ^a^	19.3%	911 ^a^	19.5%	266 ^b^	11.0%	76 ^b^	10.0%	116.2	3	<0.001	0.112
Age (Years)												V
Young	20 ^a^	4.2%	42 ^a,b^	3.0%	28 ^a,b^	2.3%	13 ^b^	1.6%	9.2	3	0.027	0.049
Young adults	97 ^a,b^	10.4%	283 ^b^	11.1%	139 ^a,c^	7.5%	50 ^c^	6.1%	27.9	3	<0.001	0.067
Older adults	289 ^a^	33.0%	996 ^a^	31.9%	334 ^b^	23.2%	125 ^b^	24.3%	47.8	3	<0.001	0.090
Older	132 ^a^	53.2%	487 ^a^	49.1%	160 ^b^	41.1%	33 ^a,b^	41.3%	11.9	3	0.008	0.083
BMI (kg/m^2^)												V
<18.5	3 ^a^	5.1%	4 ^a^	2.1%	2 ^a^	1.7%	4 ^a^	8.5%	6.7	3	0.083	0.127
[18.5–25)	85 ^a^	9.4%	367 ^a^	11.6%	163 ^b^	6.5%	49 ^c^	4.1%	80.5	3	<0.001	0.102
[25–30)	173 ^a^	20.4%	750 ^b^	25.7%	306 ^a,c^	18.3%	111 ^c^	14.8%	61.8	3	<0.001	0.100
≥30	238 ^a,b^	39.9%	625 ^b^	41.3%	178 ^a,c^	33.5%	53 ^c^	26.5%	23.3	3	<0.001	0.091

*n*, participants; %, percentage; *χ*^2^, Pearson’s chi square; df, degree freedom; *p*, *p*-value; V, Cramer’s V. Effect size; Phi, Phi’s coefficient effect size; ^a,b,c^, different letters indicate significant differences between the proportions of people with hypertension, according to sex, age, body mass index (BMI) and physical activity at 95% z-test for independent proportions.

**Table 5 healthcare-10-01283-t005:** Prevalence of use of antihypertensives by physical activity level and: sex, age and BMI.

	Physical Activity Levels											
Variables	Inactive		Walkers		Active		Very Active					ES
Sex	*n* (%)		*n* (%)		*n* (%)		*n* (%)		*χ* ^2^	df	*p*	φ
Male	242 ^a^	20.6%	751 ^a^	22.2%	314 ^b^	12.8%	93 ^c^	6.4%	226.9	3	<0.001	0.164
Female	212 ^a^	15.6%	723 ^a^	15.5%	202 ^b^	8.3%	54 ^b^	7.1%	104.5	3	<0.001	0.106
Age (Years)												V
Young	6 ^a^	1.3%	9 ^a,b^	0.7%	1 ^b^	0.1%	2 ^a,b^	0.2%	12.3	3	0.007	0.056
Young adults	66 ^a^	7.1%	158 ^a^	6.2%	79 ^b^	4.3%	19 ^b^	2.3%	28.5	3	<0.001	0.068
Older adults	259 ^a^	29.5%	838 ^a^	26.9%	288 ^b^	20.0%	94 ^b^	18.3%	46.9	3	<0.001	0.089
Older	123 ^a^	49.6%	469 ^a^	47.3%	148 ^b^	38.0%	32 ^a,b^	40.0%	12.6	3	0.006	0.086
BMI (kg/m^2^)												V
<18.5	2 ^a^	3.4%	4 ^a^	2.1%	0 ^a^	0.0%	2 ^a^	4.3%	4.3	3	0.228	0.102
[18.5–25)	75 ^a^	8.3%	258 ^a^	8.1%	106 ^b^	4.3%	19 ^c^	1.6%	89.7	3	<0.001	0.108
[25–30)	142 ^a^	16.7%	629 ^b^	21.6%	251 ^a^	15.0%	77 ^c^	10.2%	67.7	3	<0.001	0.105
≥30	201 ^a,b^	33.7%	533 ^b^	35.3%	152 ^a,c^	28.6%	46 ^c^	23.0%	17.2	3	0.001	0.078

*n* (participants); % (percentage); *χ*^2^ (Pearson’s chi square); df (degree freedom); *p* (*p*-value); V (Cramer’s V. Effect size); Phi (Phi’s coefficient. Effect size); ^a,b,c^, different letters indicate significant differences between the proportions of people using antihypertensives, according to sex, age, body mass index (BMI) and physical activity at 95% z-test for independent proportions).

**Table 6 healthcare-10-01283-t006:** Risks of hypertension, according to level of physical activity.

Physical Activity Levels										
	Inactive	Walkers			Active			Very Active		
Variables		OR	CI95%		OR	CI95%		OR	CI95%	
Overall	Ref.	1.07	0.96	1.20	0.58 *	0.51	0.66	0.41 *	0.35	0.49
Sex										
Male	Ref.	1.17 *	1.00	1.37	0.62 *	0.52	0.74	0.36 *	0.29	0.45
Female	Ref.	1.01	0.87	1.18	0.51 *	0.43	0.62	0.46 *	0.35	0.61
Age Group										
Young	Ref.	0.71	0.41	1.22	0.54 *	0.30	0.96	0.37 *	0.18	0.76
Young adults	Ref.	1.07	0.84	1.37	0.70 *	0.53	0.92	0.56 *	0.39	0.80
Older adults	Ref.	0.95	0.81	1.12	0.61 *	0.51	0.74	0.65 *	0.51	0.84
Older	Ref.	0.85	0.64	1.12	0.61 *	0.45	0.85	0.62	0.37	1.03
BMI										
<18.5	Ref.	0.40	0.09	1.84	0.32	0.05	2.00	1.74	0.37	8.17
[18.5–25)	Ref.	1.26	0.98	1.62	0.67 *	0.51	0.89	0.41 *	0.29	0.59
[25–30)	Ref.	1.36 *	1.13	1.63	0.88	0.71	1.08	0.68 *	0.52	0.88
≥30	Ref.	1.06	0.88	1.29	0.76 *	0.60	0.97	0.54 *	0.38	0.78

OR, Odds ratio, OR > 1 Indicating a higher risk of reporting hypertension; CI95%, 95% confidence interval of the odds ratio; * *p*-value < 0.05; Ref., Reference; BMI, Body mass index.

**Table 7 healthcare-10-01283-t007:** Risks of use of antihypertensive drugs, according to level of physical activity.

Physical Activity Levels										
	Inactive	Walkers			Active			Very Active		
Variables		OR	CI95%		OR	CI95%		OR	CI95%	
Overall	Ref.	1.02	0.91	1.15	0.54 *	0.47	0.62	0.32 *	0.27	0.39
Sex										
Male	Ref.	1.10	0.93	1.29	0.56 *	0.47	0.68	0.26 *	0.20	0.34
Female	Ref.	0.99	0.84	1.17	0.49 *	0.40	0.60	0.41 *	0.30	0.56
Age Group										
Young	Ref.	0.51	0.18	1.44	0.06 *	0.01	0.54	0.19 *	0.04	0.97
Young adults	Ref.	0.87	0.64	1.17	0.59 *	0.42	0.82	0.31 *	0.19	0.52
Older adults	Ref.	0.88	0.74	1.03	0.60 *	0.49	0.73	0.53 *	0.41	0.70
Older	Ref.	0.91	0.69	1.20	0.62 *	0.45	0.86	0.68	0.41	1.13
BMI										
<18.5	Ref.	0.61	0.11	3.42	0.00	n.a.	n.a.	1.27	0.17	9.35
[18.5–25)	Ref.	0.98	0.75	1.28	0.49 *	0.36	0.67	0.18 *	0.11	0.30
[25–30)	Ref.	1.37	1.12	1.68	0.88	0.70	1.10	0.57 *	0.42	0.77
≥30	Ref.	1.07	0.88	1.31	0.79	0.61	1.02	0.59 *	0.41	0.85

OR, Odds ratio, OR > 1 indicating a higher risk of reporting use of antihypertensives; CI95%, 95% confidence interval of the odds ratio; * *p*-value < 0.05; n.a., not applicable; Ref., Reference. BMI, Body mass index.

**Table 8 healthcare-10-01283-t008:** Logistic binary regression model for hypertension and use of antihypertensives risk factor.

Hypertension								
	B	S.E.	Wald	df	Sig.	Exp(B)	95% C.I. for EXP(B)	
							Lower	Upper
Inactive			33.997	3	0.000			
Walker	0.053	0.065	0.683	1	0.408	1.055	0.929	1.198
Active	−0.225	0.074	9.291	1	0.002	0.799	0.691	0.923
Very Active	−0.278	0.097	8.170	1	0.004	0.757	0.626	0.916
Sex (Men)	0.214	0.046	21.901	1	0.000	1.238	1.132	1.354
Obesity			645.762	3	0.000			
Underweight	−2.203	0.295	55.726	1	0.000	0.110	0.062	0.197
Normal	−1.526	0.061	620.344	1	0.000	0.217	0.193	0.245
Overweight	−0.836	0.055	234.338	1	0.000	0.433	0.389	0.482
Age	0.084	0.002	1500.781	1	0.000	1.088	1.083	1.093
Constant	−4.945	0.135	1332.626	1	0.000	0.007		
**Use of antihypertensives**								
	B	S.E.	Wald	df	Sig.	Exp(B)	95% C.I. for EXP(B)	
							Lower	Upper
Inactive			31.586	3	0.000			
Walker	−0.040	0.072	0.310	1	0.578	0.961	0.835	1.106
Active	−0.284	0.083	11.864	1	0.001	0.752	0.640	0.885
Very Active	−0.469	0.115	16.672	1	0.000	0.626	0.500	0.784
Sex (Men)	0.283	0.051	30.671	1	0.000	1.327	1.201	1.467
Obesity			585.659	3	0.000			
Underweight	−2.396	0.374	41.065	1	0.000	0.091	0.044	0.189
Normal	−1.657	0.070	562.107	1	0.000	0.191	0.166	0.219
Overweight	−0.862	0.060	208.648	1	0.000	0.422	0.376	0.475
Age	0.114	0.003	1645.744	1	0.000	1.120	1.114	1.126
Constant	−6.830	0.173	1559.094	1	0.000	0.001		

B (unstandardised beta); SE (standard error of regression); Wald (Wald chi square test); Df (degree freedom); Sig. (statistical significance); Exp (exponential regression); C.I. (confidence interval).

## Data Availability

The microdata were obtained on the website of the Spanish Ministry of Health, Consumer Affairs, and Social Welfare: https://www.sanidad.gob.es/estadisticas/microdatos.do (accessed on 2 March 2022).

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
