# Peer review of "Risk of Hypertension and Use of Antihypertensive Drugs in the Physically Active Population under-70 Years Old—Spanish Health Survey"

_healthcare, 2022, doi:10.3390/healthcare10071283_

Round 1
Reviewer 1 Report
The paper is interesting but there is a risk for the reader of overestimating the strenght of the reported results, since the data are only derived from a survey on a self reported prevalence of hypertension and hypertensive drugs. No objective data on BP levels and types and number of antihypertensive drugs are reported. All this should be more clearly stated throughout the paper, also in the Title, in the Discussion, in the Limitation and in the Conclusions. The Conclusions should be then considered only as suggestive and/or hypothesis generating.
The sentence in the last line page 11, and first 2 lines page 12: "However, randomized controlled trials with longer intervention periods and observational studies with objectively measured PA are needed to define the exact dose–response relationship of PA and hypertension." should be also part of the Conclusions.
We suggest also to modify the Title of the paper as follows: " Reduced risk of self-reported hypertension and use of antihypertensive drugs in the physically active Spanish population below 70 years of age."
Page 2, line 12: among the reasons for the high prevalence of hypertension also genetic components and family history should be also taken into account.
Page 2, 5th paragraph, line 2: instead of "less" should be probably "higher than", both for SBP and DBP.
Pages 3 and 4: Subjets and Procedures: the questions of the survey are not clear and should be reported in more detail, probably in a separate explicative Table. It is not clear if newly diagnosed hypertension was separately classified as compared to established hypertension and if duration of hypertensive treatment was recorded. It should be also reported if the types and the number of drugs used for antihypertensive treatments, and the achieved BP reduction, were analyzed and if this could influence the results. All this should be discussed in detail in the Discussion and Limitation sections.
The questions reported at page 4, line 2 and line 7 appear very approximate and generic. This aspect should be also discussed in the Limitations. It is also not clear which BP cut-offs were considered for the definition of hypertension.
Page 4; last 3 lines before Statistical Analysis: instead of "PAI" it should be probably "PAL" (repeated 5 times).
For the Statistical Analysis why ANOVA within and between groups was not used and considered ?
Page 9, first line: instead of "an", it should be probably "and".
Page 10, line 5: how the Authors explain "except for the Very active older " ?
Among the Limitations the cut-off age of 70 years as an exclusion criterion should be also considered as a limitation.
Author Response
Reviewer 1
The paper is interesting but there is a risk for the reader of overestimating the strenght of the reported results, since the data are only derived from a survey on a self reported prevalence of hypertension and hypertensive drugs. No objective data on BP levels and types and number of antihypertensive drugs are reported. All this should be more clearly stated throughout the paper, also in the Title, in the Discussion, in the Limitation and in the Conclusions. The Conclusions should be then considered only as suggestive and/or hypothesis generating.
Authors - Dear reviewer, we appreciate the comments and suggestions made, which greatly contribute to improve the quality of the manuscript. We have amended it according to the indications and have marked it in the track text of the mentioned sections.
The sentence in the last line page 11, and first 2 lines page 12: "However, randomized controlled trials with longer intervention periods and observational studies with objectively measured PA are needed to define the exact dose–response relationship of PA and hypertension." should be also part of the Conclusions.
Authors - Dear reviewer, we appreciate the comment, and we have amended it accordingly. To avoid repetition, sentence in conclusion was changed to the following: “Nonetheless, controlled trials with longitudinal interventions and randomized observational studies with objective measures of PA are needed to characterize the exact dose–response relationship between PA and hypertension.
We suggest also to modify the Title of the paper as follows: " Reduced risk of self-reported hypertension and use of antihypertensive drugs in the physically active Spanish population below 70 years of age."
Authors – Thank you, we proceeded accordingly. Considering that other reviewer also suggested related to the title, the following update was made: “Risk of hypertension and use of antihypertensive drugs in the physically active population under- 70 years old - Spanish Health Survey”
Page 2, line 12: among the reasons for the high prevalence of hypertension also genetic components and family history should be also taken into account.
Authors – Thank you, we proceeded accordingly and the changes are highlighted in the text
Page 2, 5th paragraph, line 2: instead of "less" should be probably "higher than", both for SBP and DBP.
Authors – Thank you for the amendment, we corrected accordingly
Pages 3 and 4: Subjets and Procedures: the questions of the survey are not clear and should be reported in more detail, probably in a separate explicative Table. It is not clear if newly diagnosed hypertension was separately classified as compared to established hypertension and if duration of hypertensive treatment was recorded. It should be also reported if the types and the number of drugs used for antihypertensive treatments, and the achieved BP reduction, were analyzed and if this could influence the results. All this should be discussed in detail in the Discussion and Limitation sections.
Authors – Thank you, we proceeded accordingly and the changes are highlighted in the manuscript. In order to address your concerns, we have also added Table 1 in the Procedures section.
The questions reported at page 4, line 2 and line 7 appear very approximate and generic. This aspect should be also discussed in the Limitations. It is also not clear which BP cut-offs were considered for the definition of hypertension.
Authors – Thank you, we proceeded accordingly. We have clarified and added the information and the changes are highlighted in the manuscript
Page 4; last 3 lines before Statistical Analysis: instead of "PAI" it should be probably "PAL" (repeated 5 times).
Authors – Thank you for your comment. We have modified this section to facilitate understanding of the variables and procedures followed.
For the Statistical Analysis why ANOVA within and between groups was not used and considered?
Authors – We consider that an ANOVA could not be performed, since the data did not follow a normal distribution. The analyses were performed, analysed relative frequencies, so a pairwise z-test for independent proportions was chosen, whose philosophy is similar to the Bonferroni correction of an ANOVA, although for proportions. Additionally, we have added data from multiple binary logistic regression analyses, using as dependent variables: hypertension status and antihypertensive use; respectively, and age, BMI group and PAL, as independent variables, which we believe has increased the quality of our manuscript.
Page 9, first line: instead of "an", it should be probably "and".
Authors – Fixed. Thank you
Page 10, line 5: how the Authors explain "except for the Very active older " ?
Among the Limitations the cut-off age of 70 years as an exclusion criterion should be also considered as a limitation.
Authors – Thank you, we proceeded accordingly and the changes are highlighted in the manuscript
Reviewer 2 Report
This study provides findings using specimens from the Spanish national survey and is considered to be at a level that can be adopted. However, the following points are extremely inadequate and need improvement.
1. "Reduced" in the title and "reduced" (for example, page 11, line 33) and "reduce" (for example, page 11, line 37) are often used in the text. Careful consideration is required for their use. Because this study is a cross-sectional study, it cannot be said that the result is that the frequency of hypertension etc. is low due to high level of physical activity. I may have started to work hard because of high blood pressure. In any case, cross-sectional studies should remove any confusing notations that could be perceived as causal. The parts that need to be changed are as follows.
"Reduced risk" in the title "Reduced prevalence" on page 11, line 33 “Seems to reduce” on page 11, line 37 "The reduction of the prevalence" on page 11, line 44 “Had reduced risks” on page 12, line 18 “Showed reduced risks” on page 13, line 11 "PA could reduce" on page 13, line 13
2. There is an expression "SNHS 2017" in Tables 1 and 2, but is it correct? The analysis target of this study started from 23089 people of SNHS 2017, but since the analysis target has decreased to 17964 people due to lack of hypertension data etc., it is possible for readers to misunderstand that it is purely the analysis result from SNHS 2017. High sex. It should be deleted or corrected to the appropriate wording.
3. This is not necessarily an additional analysis, but I thought it might be possible to do a similar analysis with either hypertension or taking medication for hypertension as the dependent variable. I think that what the author originally wants to analyze will not be clear without this analysis.
4. I think it's just a careless mistake, but the year of publication in the margin is "2021". I think it will be corrected at the time of publication, but it is confusing, so please correct it.
Author Response
Reviewer 2
This study provides findings using specimens from the Spanish national survey and is considered to be at a level that can be adopted. However, the following points are extremely inadequate and need improvement.
- "Reduced" in the title and "reduced" (for example, page 11, line 33) and "reduce" (for example, page 11, line 37) are often used in the text. Careful consideration is required for their use. Because this study is a cross-sectional study, it cannot be said that the result is that the frequency of hypertension etc. is low due to high level of physical activity. I may have started to work hard because of high blood pressure. In any case, cross-sectional studies should remove any confusing notations that could be perceived as causal. The parts that need to be changed are as follows.
"Reduced risk" in the title "Reduced prevalence" on page 11, line 33 “Seems to reduce” on page 11, line 37 "The reduction of the prevalence" on page 11, line 44 “Had reduced risks” on page 12, line 18 “Showed reduced risks” on page 13, line 11 "PA could reduce" on page 13, line 13
Authors - Dear reviewer, we appreciate the comments and suggestions made, which greatly contribute to improving the quality of the manuscript. We have amended it according to the indications and have marked it in the track text of the mentioned section
- There is an expression "SNHS 2017" in Tables 1 and 2, but is it correct? The analysis target of this study started from 23089 people of SNHS 2017, but since the analysis target has decreased to 17964 people due to lack of hypertension data etc., it is possible for readers to misunderstand that it is purely the analysis result from SNHS 2017. High sex. It should be deleted or corrected to the appropriate wording.
Authors - Thank you for your appreciation, the titles of the tables have been corrected.
- This is not necessarily an additional analysis, but I thought it might be possible to do a similar analysis with either hypertension or taking medication for hypertension as the dependent variable.I think that what the author originally wants to analyze will not be clear without this analysis.
Authors - Thank you for your suggestion. We have added two binary multiple logistic regression analyses, using hypertension status and antihypertensive use as dependent variables, respectively, with age, BMI group and PAL as independent variables.
- I think it's just a careless mistake, but the year of publication in the margin is "2021".I think it will be corrected at the time of publication, but it is confusing, so please correct it.
Authors - Corrected, thank you.
Reviewer 3 Report
I congratulate the authors for this fascinating paper, which sheds light on the efficacy of PA on BP control.
Here are my personal comments and suggestions:
1. Section Subjects: Please insert the full questions of Q.25a.1 and the others.
2. Section Procedures: what do you consider Vigorous and moderate PA? Have you provided examples for the subjects that respond to your survey?
If it is affirmative, please insert the definition or the examples of Vigourous and moderate PA.
3. Section Results: to exclude the BMI, sex and age effect on PAL and hypertension and the use of antihypertensive drugs, the authors could perform a multivariate analysis, creating 3 models for exclusion of sex, BMI and age, respectively.
In this way, the effects of PAL and hypertension may be investigated better.
Best regards.
Author Response
Reviewer 3
I congratulate the authors for this fascinating paper, which sheds light on the efficacy of PA on BP control.
Authors - Dear reviewer, we appreciate the comments and suggestions made, which greatly contribute to improving the quality of the manuscript. We have amended it according to the indications and have marked it in the track text of the mentioned sections
Here are my personal com
- Section Subjects: Please insert the full questions of Q.25a.1 and the others.
Authors – Thank you, we proceeded accordingly and the changes are highlighted in the manuscript
- Section Procedures: what do you consider Vigorous and moderate PA? Have you provided examples for the subjects that respond to your survey?
If it is affirmative, please insert the definition or the examples of Vigourous and moderate PA.
Authors – Thank you question. In order to answer it, we have added a table with the description of the mentioned points in the procedures section. Thank you
- Section Results: to exclude the BMI, sex and age effect on PAL and hypertension and the use of antihypertensive drugs, the authors could perform a multivariate analysis, creating 3 models for exclusion of sex, BMI and age, respectively.
In this way, the effects of PAL and hypertension may be investigated better.
Authors - Thank you for your suggestions. Due to the nature of our dependent variables (dichotomous categorical variables) and the distributions followed by the data (no evidence to assume normality) we considered that we could not use this type of analysis. However, we have included a multiple binary logistic regression analysis, with hypertension status and antihypertensive use, respectively, as dependent variables, and age, BMI group and PAL as independent variables. We believe that our manuscript has a more effective presentation of the results.
Best regards.
Round 2
Reviewer 1 Report
In the abstract: " seems to be reduced.. "
Reviewer 2 Report
I judged that this manuscript has been properly modified and can be published.
Reviewer 3 Report
In this new version of the manuscript, the new evidence and the binary analysis complete the framework of the work. All suggestions are taken into consideration. The manuscript now is ready to be published.